# Morphogenetic metasurfaces: unlocking the potential of Turing patterns

Thomas Fromenteze [1] ✉, Okan Yurduseven[2], Chidinma Uche[1], Eric Arnaud[1], David R. Smith[3] & Cyril Decroze [1]

The reaction-diffusion principle imagined by Alan Turing in an attempt to explain the structuring of living organisms is leveraged in this work for the procedural synthesis of radiating metasurfaces. The adaptation of this morphogenesis technique ensures the growth of anisotropic cellular patterns automatically arranged to satisfy local electromagnetic constraints, facilitating the radiation of waves controlled in frequency, space, and polarization. Experimental validations of this method are presented, designing morphogenetic metasurfaces radiating far-field circularly polarized beams and generating a polarization-multiplexed hologram in the radiative near-field zone. The exploitation of morphogenesis-inspired models proves particularly well suited for solving generative design problems, converting global physical constraints into local interactions of simulated chemical reactants ensuring the emergence of self-organizing meta-atoms.

Alan Turing's mathematical framework for understanding morphogenesis, the process by which complex structures in living organisms arise, has proven effective at reproducing the spontaneous formation of numerous biological patterns[1–4]. He considered the diffusion of antagonistic chemical species that he named *morphogens*, reacting according to their respective concentrations to form diverse patterns, following predator-prey dynamics[5].

The effectiveness of such models for the synthesis of biological characteristics and functions relies on the decentralization of the instructions guiding the structuring of life[6]. This need is easily justified by the absence of any internal organ ensuring a decision-making role during the early stages of development, thus distributed through local interaction mechanisms. R. Dousart et al. have conceptualized the development of such morphogenesis-inspired generation techniques based on self-organizing structures to simplify the engineering of complex systems[7–9]. Faced with increasingly sophisticated specifications, the latter essentially rely on combination of independent components whose design is concentrated in the hands of a few specialists. Through the development of morphogenetic engineering, it is thus proposed to draw inspiration from the self-structuring capacity of living organisms to ensure the synthesis of entire systems,

decentralizing the design burden by defining simplified rules of generation at a local scale. This emergent domain has so far been the subject of only a limited number of applications outside the field of biology, including the distributed control of swarms of robots[10], the advanced design of agent-based models[11] and the topological optimization of mechanical structures[12].

In this work, we present a gradient-descent free generative model of morphogenetic electromagnetic design adapted to the automated synthesis of radiating metasurfaces. The interaction mechanisms imagined by A. Turing here guide the emergence and structuring of patterns according to a unique generation process to synthesize the desired electromagnetic properties. The Gray-Scott model[13,14] is often retained among the great variety of studied reaction-diffusion systems (Fig. 1), taking advantage of extensive pattern classification efforts[15,16]. Nevertheless, it remains necessary to adapt this model to the specific constraints of the considered application domain. The vector nature of electromagnetic waves indeed requires a control of anisotropic properties. A supplementary degree of freedom is thus exploited in this work by implementing direction-selective diffusion mechanisms of the morphogens, facilitating the synthesis of tensorial electromagnetic constraints. Such approaches have been introduced in the

[1]University of Limoges, CNRS, XLIM, UMR 7252, F-87000 Limoges, France. [2]Centre for Wireless Innovation (CWI), Institute of Electronics, Communications and Information Technology (ECIT), Queen's University Belfast, Belfast BT3 9DT, UK. [3]Center for Metamaterials and Integrated Plasmonics, Department of Electrical and Computer Engineering, Duke University, Durham, NC 27708, USA. ✉e-mail: thomas.fromenteze@unilim.fr

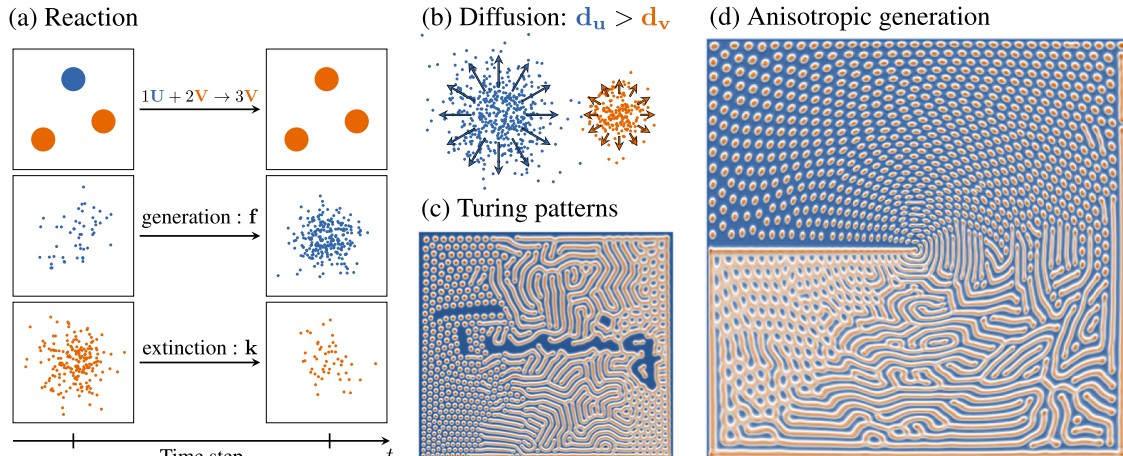

**Fig. 1 | Constrained generation of anisotropic Turing patterns.** The Gray-Scott model is based on the simulation of virtual, antagonistic chemical compounds referred to by A. Turing as "morphogens". Following simple reaction (**a**) and diffusion (**b**) mechanisms, these morphogens, separated into two categories $U$ and $V$, form a system of preys and predators, providing the opportunity to establish population equilibria generating spatial patterns, known as Turing patterns (**c**). The diffusion constants $d_u$ and $d_v$ control the dimensions of the patterns generated, while the pair of parameters $f$ and $k$ influence their type. This model is modified to offer a new degree of freedom with anisotropic characteristics (**d**). Each point in space is thus associated with a diffusion tensor whose eigenaxes determine preferred directions influencing the local orientation of the synthesized patterns.

field of computer graphics for the generation of textures[17–19] and have notably been adapted to the visualization of vector fields[20] and to the generation of microfluidic components[21]. By proposing a method for the automated synthesis of anisotropic electromagnetic characteristics, it is demonstrated in this work that it is possible to automate the generation of metasurfaces meeting complex and polarization-constrained radiation objectives.

## Results

### Emergence of self-organized anisotropic patterns

The generative model adapted for the automated synthesis of metasurfaces is summarized in Fig. 1.

Following the mechanisms studied by A. Turing, the Gray-Scott model simulates the evolution of a dynamical system where the chemical species **U** and **V** diffuse in space according to the coefficients **d$_u$** and **d$_v$**. In the presence of one $U$-morphogen and two $V$-morphogens, the latter is converted into a third $V$. The populations of **U** and **V** evolve according to the spontaneous generation coefficient **f** (feed) for the $U$ morphogens and spontaneous extinction **k** (kill) for the $V$ morphogens. The emergence of patterns is finally associated with pairs of parameters **f** and **k**, corresponding to population equilibria. Following an Eulerian discretization model, only the concentration of the morphogens is considered to simplify the computations. The finite difference resolution of this mathematical model thus simulates the growth of Turing patterns, occupying all available space (Fig. 1-C). The local control of the orientation of the generated patterns is ensured in this work with the use of anisotropic Laplacian operators. To this end, Diffusion tensors **D** are defined at each point of space at a scale smaller than the generated patterns, adapting the ratio of their eigenaxes to induce direction-selective diffusions of morphogens.

Morphogenesis represents a source of inspiration for guiding electromagnetic design, focusing more specifically in this work on the procedural synthesis of antenna metasurfaces. These antennas operate on the principle of leaky waveguides[22], requiring the definition of a spatial modulation of the electromagnetic characteristics of a plane so that incident surface waves can be converted into radiated ones[23–26]. The objective of this work is to demonstrate the possible procedural synthesis by a morphogenetic technique of metasurfaces whose radiation can be constrained in space, in frequency and in polarization. Compared to existing microwave and photonic techniques where

elements are generated independently on periodic (essentially Cartesian) grids, self-replicating spots here automatically settle in space, rapidly converging to compact spatial arrangements without uniformity constraints in order to reach the radiation objectives. The patterns thus migrate to obtain a denser packing with regular spacings regardless of their orientation[27]. The introduction of a partial disorder also limits the amplification of diffusion effects along the main axes of the crystals formed by periodic structures, facilitating the homogenization of the synthesized physical properties[28,29].

This work falls within a context where the design automation of complex electromagnetic systems is the subject of increasingly advanced proofs of concept, in particular proposed within the framework of generative design activities[30–32].

Exploiting mechanisms inspired by morphogenesis, the conversion of macroscopic electromagnetic objectives is here directly converted into local rules guiding the generation of the exploited structures. This decentralization of the design burden associated with the self-structuring properties of the proposed generative model restricts the electromagnetic engineering process to a single definition of environmental variables guiding morphogenesis, making it gradient-descent free and natively scalable to large problems.

### Procedural generation of morphogenetic metasurfaces

Considering the generative model introduced in Fig. 1, one must ensure the conversion of the objective electromagnetic characteristics into morphogenetic parameters (Fig. 2).

The latter will then guide the growth of Turing patterns, etched on the upper surface of a parallel plate waveguide formed by a dielectric substrate (Rogers 3003 · $\epsilon_r = 3$, thickness = 1.524 mm) arranged between two thin copper sheets. The definition of the electromagnetic characteristics is realized by considering the interaction between a dominant transverse magnetic surface wave excited at the center of the circuit by a monopole element and an objective electric radiation back-propagated onto the metasurface.

Depending on a set of steps summarized in the Method section and developed in section 1 of the supplementary materials, a modulation of the imaginary part of the surface impedance (the reactance) ensures an efficient conversion between the excited surface wave and objective leaky waves[24]. The control of reactance tensors requires first a conversion of morphogenetic parameters into electromagnetic

(a)

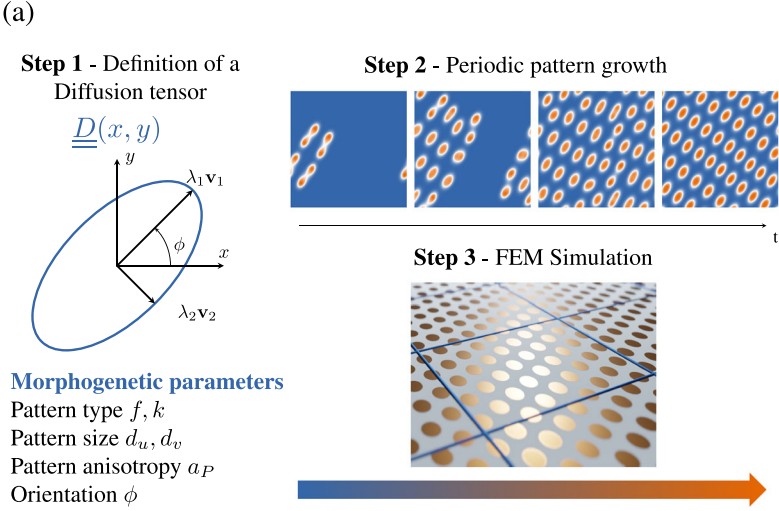

**Step 1** - Definition of a Diffusion tensor

$\underline{D}(x,y)$

**Morphogenetic parameters**
Pattern type $f, k$
Pattern size $d_u, d_v$
Pattern anisotropy $a_P$
Orientation $\phi$

**Step 2** - Periodic pattern growth

**Step 3** - FEM Simulation

**Step 4** - Characterization of the associated reactance tensor

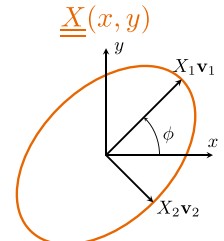

$\underline{X}(x,y)$

**Electromagnetic parameters**
Average reactance $X_{sw}$
Reactane anisotropy $a_X$
Orientation $\phi$

(b)

**Step 1** - Objective reactance tensor definition

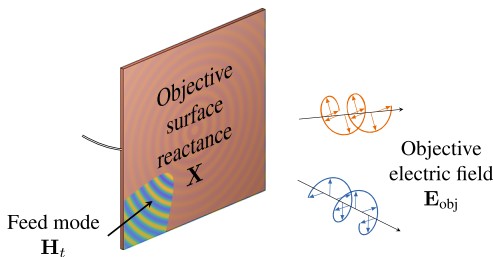

Objective surface reactance **X**

Feed mode $\mathbf{H}_t$

Objective electric field $\mathbf{E}_{obj}$

**Step 2** - Metasurface generation

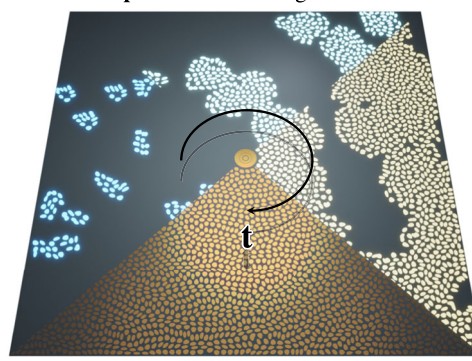

**Fig. 2 | Morphogenetic metasurface characterization and synthesis.**
**a** Characterization of the surface reactance controlled by morphogenetic parameters: A first characterization phase enables generation parameters to be associated with the electromagnetic properties synthesized. The study is restricted to the growth of cellular patterns whose ellipticity, dimensions and orientation are controlled by local constraints influencing morphogen self-structuring. The parameters are identical in space, which is defined with periodic boundary conditions to obtain spatial distributions enabling continuous tessellation of the plane. Finite element simulation is used to extract the electromagnetic characteristics associated with each set of morphogenetic parameters. Numerical processing is then applied to derive the reactance tensors required to convert surface waves into radiated waves. **b** Morphogenetic metasurface synthesis: the definition of radiation objectives enables the desired electromagnetic characteristics to be derived at the surface of a parallel plate waveguide excited by a monopole. Conversion of these targets into local morphogenetic parameters is required for metasurface generation. An iterative resolution of the anisotropic Gray-Scott model ensures the growth of elliptical patterns undergoing a series of successive divisions until they occupy the entire permitted space. Under the influence of the morphogenetic parameters, the Turing patterns self-structure to synthesize the desired electromagnetic tensors.

properties. Among the Turing patterns synthesizable with this generative model, we restrict ourselves in this work to the pairs of parameters **f** and **k** triggering the synthesis of cellular elements. These patterns exploit a self-duplication mechanism similar to mitosis, occupying space rapidly by a series of successive divisions. By reproducing many features associated to the development of living organisms, such as dissipation, autocatalysis and homeostasis[33], the emergence and self-organization of spatial structures here guide the synthesis of desired properties.

Figure 2 depicts the steps involved in the characterization of the generated patterns using finite element method (FEM) simulations, successively varying the ratio of the eigenaxes of the diffusion tensors used for the generation to control the anisotropy of the associated reactance tensors. Ensuring the growth of cellular elements whose periodicity is smaller than the operating wavelength, it is possible to extract electromagnetic properties by homogenization. The conversion of morphogenetic parameters into reactance tensors is finally achieved by following the numerical processing detailed in section 2 of the supplementary materials. We noticed that the orientation of the eigenaxes of the diffusion tensors and the reactance tensors remain identical, reducing the characterization to modulation of the ratio of the eigenaxes of the diffusion tensors. These simulations rely on the generation of spatially periodic patterns, provided natively through to the implementation of discrete Laplacian operators by finite difference.

## Synthesis of far-field radiation objectives
A first experimental validation of this generative model is proposed by the synthesis of radiating metasurfaces constrained in far-field. This design automation addresses a crucial issue in telecommunication, where the optimization of the energy radiated in desired directions and polarization states directly impacts the performance of data transfers and generally justifies the intervention of experienced antenna engineers. A performance metric of realized gain is thus considered, corresponding to the power density in the beam direction of an antenna for a given polarization, divided by the power density of an equivalent isotropic antenna. This characteristic considers all-dielectric, ohmic and impedance mismatch losses.

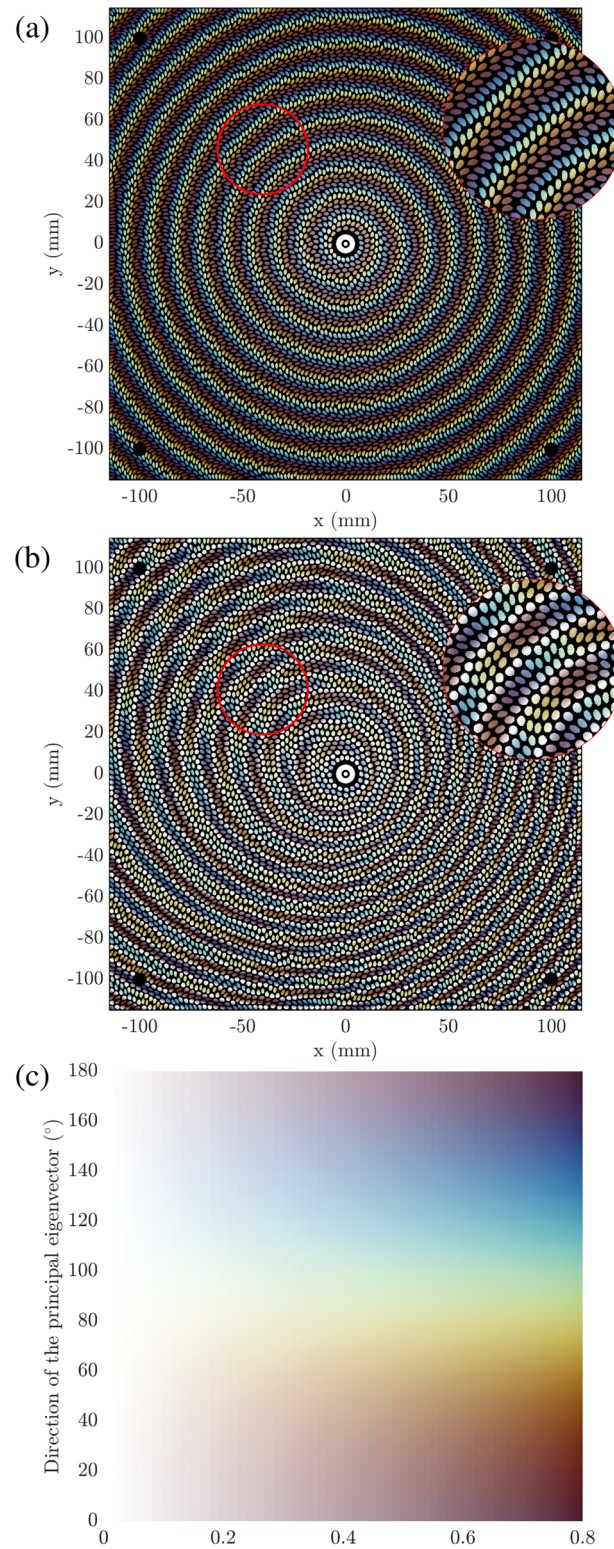

**Fig. 3 | Demonstrations of procedural generation of morphogenetic metasurfaces. a** Metasurface radiating a broadside LHCP beam. **b** Dual-polarized metasurface radiating LHCP and RHCP beams simultaneously in two different directions. **c** Colormap representing orientation and anisotropy of objective tensors.

Following the procedure presented in Fig. 2, it is now possible to convert radiation constraints into local morphogenetic parameters guiding the emergence of elliptical patterns, self-structuring to satisfy the expected electromagnetic objectives. Geometrical boundary conditions are also directly imposed by the definition environmental variables, prohibiting the reproduction of cells in specific areas by locally increasing the extinction rate **k**. The metasurfaces designed for these demonstrations are of square shape with 229 mm sides (Fig. 3). In these structures, the growth of morphogens is forbidden in the center of the metasurfaces in a radius of 7 mm to facilitate the design of a common impedance matching structure enhancing the coupling of surface waves. Four circular areas of 5mm diameter placed at the extremities of the metasurfaces are also constrained in the same way, leaving the space available for the insertion of nylon screws. Among the objectives considered, it seems interesting to demonstrate the ability of this generation technique to adapt to various boundary conditions, maximizing the exploitation of the entire footprint of each radiating surface without having to increase their dimensions to ensure mounting.

For this demonstration, two morphogenetic metasurfaces are synthesized to radiate far-field circularly polarized beams (Fig. 3). A first antenna is designed to maximize the energy radiated in the axis perpendicular to its plane in left-hand circular polarization while maintaining a low level of cross-polarization at 20 GHz. A second antenna is designed to radiate at the same frequency two beams steered of 30° in elevation and − 30° in azimuth, respectively left and right-hand circularly polarized. Animations of procedural generations of single-polarized and dual-polarized metasurfaces are available in the supplementary materials (Supplementary Movie 1 and 2) and the experimental results are presented in Fig. 4 and in Tab. 1.

The measured performances are in general good agreement with the values determined by simulation in both cases, computed using the time domain solver of the commercial software CST Studio Suite. These measurement results reveal a low level of losses, reflected in the differences between gain and directivity measurements, ranging from 0.4 dB to 0.7 dB. These characteristics are guaranteed by the simple operation of these metasurfaces, avoiding the use of complex circuits for the feeding of the radiating elements[34]. The synthesis of the two metasurfaces was constrained to ensure a maximum gain at 20 GHz. The measurements revealed a slight frequency drift, between 1% and 1.5%, attributable to the sensitivity of the synthesized reactance properties to the dispersion of the dielectric properties of the substrate ($\epsilon_r = 3 \pm 0.04$), as well as to the simplifications of the models developed in section 1 of the supplementary materials. The frequency measurements developed in these same appendices also attest to the weak impact of these dispersions at 20 GHz.

The aperture efficiency, defined as the ratio of the effective radiating area to the actual physical aperture, is determined for this first metasurface at 49%. Considering the simplicity of the electromagnetic models used, this result is in good agreement with the characteristics presented in the scientific literature using conventional design methods[35,36]. However, these contributions are based on essentially circular shapes to limit the impact of the small contribution of the substrate angles to the radiation and to this performance metric, also developing decomposition formalisms specifically adapted to these geometries to improve the definition of the electromagnetic characteristics to be synthesized[37,38]. Exploiting the whole footprint of the metasurfaces and including constraints facilitating their mounting, it is thus possible to demonstrate the adaptation of this morphogenetic technique to the synthesis of freeform metasurfaces[39].

As expected, the second measured morphogenetic metasurface radiates simultaneously two orthogonal circular polarizations in two different directions. The measurements are presented for the maximum gains of each beam and confronted to simulation results (Fig. 4). The maximum power densities are radiated in a direction of −29.5° in azimuth for the left-hand circular polarization and 29.5° in elevation for right-hand circular polarization, instead of the expected −30° in azimuth and 30° in elevation. These slight angular drifts are justified by the previously mentioned weak frequency dispersion. Beyond such

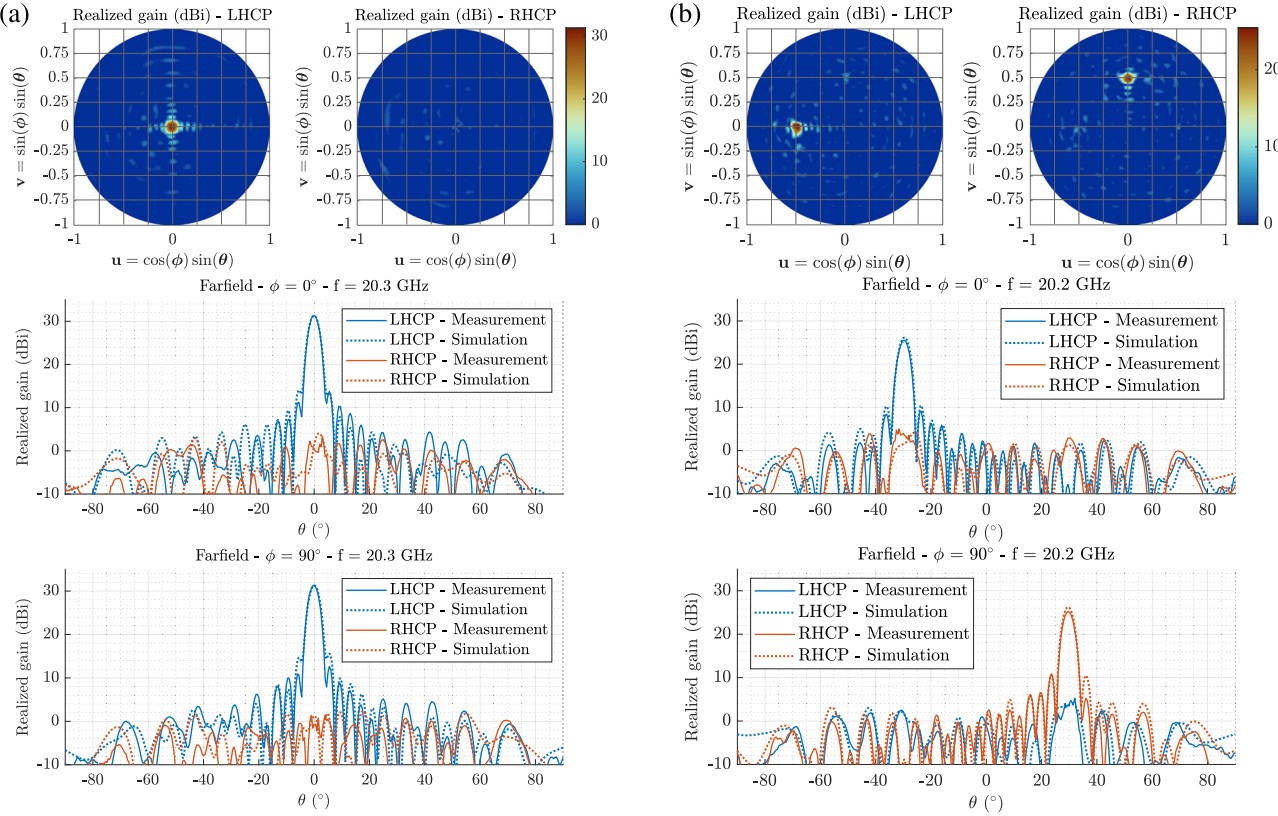

**Fig. 4 | Experimental validations of two morphogenetic metasurfaces measured in far-field.** The realized gains are measured at 20.3GHz for the single-polarized (**a**) and dual-polarized (**b**) metasurfaces and presented according to orthographic projections and for two cutting planes.

**Table 1 | Synthesis of the experimental results of the two far-field radiating morphogenetic metasurfaces**

| Metasurface Antenna | Polarization | Gain (dBi) | Directivity (dBi) | Frequency drift (%) | Aperture efficiency (%) |
|---|---|---|---|---|---|
| Single Polarized | LHCP | 31.3 [31.3] | 31.7 | 1.5 | 49 |
| Dual Polarized | LHCP | 25.4 [26.2] | 26.1 | 1 | 30 |
| | RHCP | 25.2 [26.2] | 25.9 | 1 | 31 |

The gain values in square brackets correspond to the associated simulation results.

minor deviations, these measurements attest to the good adaptation of the generative model to simultaneous radiation objectives, fitting the environmental variables to linear combinations of characteristics to be synthesized. The gains measured for the dual-polarized metasurface are naturally lower than for the first antenna, considering a reduction by two of the effective area allocated to each radiation objective. Moreover, the beam steering introduces an additional limitation of the effective radiating apertures by a factor $\cos(30°)$. In spite of these constrained performances, the obtained results remain in good agreement with numerical predictions and demonstrate the self-organization capacity of the meta-atoms according to the only local rules defined on each metasurface.

### Polarization-multiplexed near-field holography
A third morphogenetic metasurface is this time synthesized with radiation constraints imposed in the radiated near-field region. To demonstrate the ability of the generated patterns to satisfy more sophisticated holographic objectives, it is this time required to control the radiated field so as to form the letters "L" and "R", respectively circularly polarized to the left and to the right. This field constraint is imposed in a parallel plane set at 20 cm from the metasurface (Fig. 5). The measured near-field, from which the results are extracted at 20.2 GHz (following a frequency drift of 1% consistent with previous

demonstrations), reveals spatial distributions corresponding to the expected shapes while demonstrating a weak cross-polarization between the latter. An animation of the measured field is provided in the Supplementary Movie 3.

The scientific literature already includes several experiments of near-field focusing techniques using metasurfaces or analogous and older principles of leaky-wave antennas[40–42]. The demonstration proposed in this work is, however, distinguished by the complexity of the generated hologram, whose level of spatial and polarization control has to the authors' knowledge, never been achieved in the microwave range with such a simple radiating structure.

This demonstration provides further evidence of the ability of the generated patterns to spatially self-structure to fit the electromagnetic constraints imposed on the metasurface. Without the definition of regular layout grids, elliptical cells can migrate and deform under the action of local morphogenetic parameters to form a compact and complex arrangement synthesizing the expected spatially varying anisotropic properties.

### Discussion
The presented experimental validations have highlighted the efficiency and flexibility of the considered morphogenetic model, proposed in the framework of first-order approximations to facilitate both

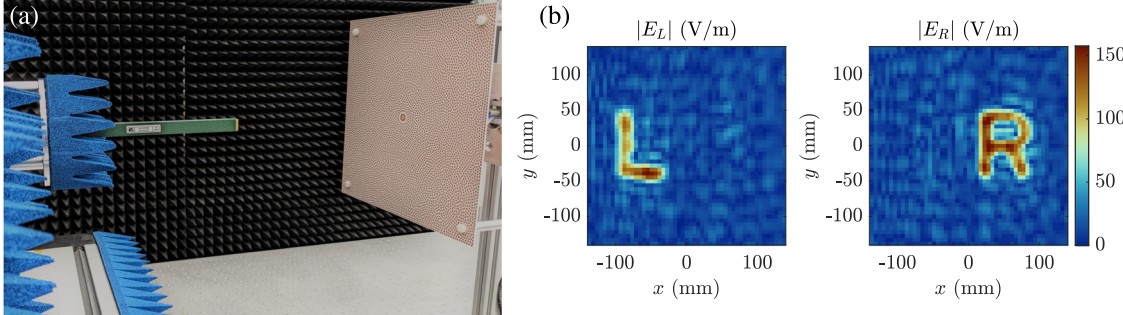

**Fig. 5 | Experimental demonstration of a polarization-multiplexed hologram at 20.2 GHz. a** Field scan of a morphogenetic metasurface. **b** Measured electric field distributions represented according to a left-hand circular polarization ($|E_L|$) and a right-hand circular polarization ($|E_R|$).

the description of these proofs of concept and their reproductions. The main originality of this work lies in the definition of a generative technique facilitating the decentralization of the design burden[8]. The proposed model thus facilitates the determination of a set of particularly simple interaction rules at a local scale, deduced from electromagnetic characteristics defined for the whole synthesized component and providing an highly scalable method. The optimization of the performances achievable by means of more complex electromagnetic formulations[39,43] nevertheless leaves possible margins of progress open in the wake of this first demonstration. The spatial self-structuring properties of the proposed technique enable more densely and uniformly arranged spatial distributions to be generated, compared with conventional methods based on periodic arrangements of meta-atoms. Some advantages linked to these properties are presented in section 7 of the supplementary materials, but in-depth studies will be needed to fully quantify the benefits of this technique. Although these demonstrations are presented in the microwave range, a direct transposition of the proposed techniques is of course conceivable in the photonic domain, adapting the reference wave and radiation objectives as required. The self-structuring capability of meta-atoms made of dielectric materials would offer a potential answer to the limitations associated with periodicity and geometric breaks in uniform lattices, enabling the fundamental limits of conventional metasurfaces to be re-explored[44]. Recent work has shown that these techniques can be used to synthesize partially correlated disordered media[45]. Eliminating the need for axes of symmetry by introducing controlled disorder would, via the proposed morphogenetic technique, provide a means of conceiving designs that could be more robust to fabrication errors[46].

Beyond the considered electromagnetic application, many perspectives of adaptation of this work to automated design in various fields of physics and engineering can be envisioned. In each case, the generation requires a single phase of characterization of the synthesized properties according to morphogenetic parameters. The process can then be reversed, deducing from macroscopic objectives the parameter sets guiding the morphogenesis of complex architectures.

These research activities finally fall within a broader context of the development of artificial life[47]. Identifying fundamental mechanisms exploited by living organisms, these advances open the way to the design of dynamic systems governed by complex interactions and a capacity for self-structuring to ensure the synthesis of physical characteristics and functions. In such a context, the adaptation of associative learning mechanisms based on the encoding of environmental information[48] and the definition of sophisticated interaction rules using differentiable programming techniques[49] are expected to open particularly promising perspectives for the elaboration of more efficient and complex generative models.

## Methods

The following subsections gather a synthesis of the steps considered for the development and validation of this work. For the sake of compactness, the elements presented are a condensed version of the complete and illustrated developments provided in sections 1, 2, and 3 of the supplementary materials.

### Definition of objective reactance tensors

The generation of metasurfaces must be constrained by the definition of objective electromagnetic parameters. A vertical monopole placed in the center of this structure radiates a dominantly transverse magnetic wave, guided between two copper plates within a dielectric substrate. The etching of patterns on the upper surface enables the conversion of surface waves into radiated waves by means of reactance tensors that must be modulated over the entire metasurface. A first-order electromagnetic model is considered, allowing both to facilitate the reproducibility of this work and to demonstrate the effectiveness of the proposed approach despite the use of simplified formalisms. In this approximation, the surface waves are weakly perturbed by the reactance modulation :

$$\mathbf{E}_t|_{z=0^+} = j\underline{\underline{\mathbf{X}}} \cdot \mathbf{J} = j\underline{\underline{\mathbf{X}}} \cdot J_0\, H_1^{(2)}(-jk_{sw}\rho)\hat{\boldsymbol{\rho}}, \quad (1)$$

The wave vector $k_{sw} \approx j\beta_{sw}$ reflects by its imaginary part the average phase delay cumulated per unit length by the incident wave. This last quantity depends directly on the average surface reactance $X_{sw}$ exhibited by the printed patterns:

$$\beta_{sw} = k_0\sqrt{1 + \left(\frac{X_{sw}}{\eta_0}\right)^2}, \quad (2)$$

where $k_0$ and $\eta_0$ represent, respectively the wavenumber and the free space impedance. For this work, the patterns will be generated to maintain a constant average reactance $X_{sw}$, thus determining the spatial frequencies of the printed patterns and guaranteeing the adaptation of the monopole exciting the reference wave. A holographic approach is considered in order to facilitate the definition of reactance tensors[24]. Following Eq. (1), the objective is thus to synthesize a surface impedance able to transform a reference wave, corresponding here to the currents $\mathbf{J}$, into a radiated wave, defined just above the surface by $\mathbf{E}_t|_{z=0^+}$. The reactance tensor to be synthesized being by definition of higher rank than the transformed vectors, there is no unique definition of its components and additional constraints can be considered to facilitate its derivation. First of all, the reactance tensors $\underline{\underline{\mathbf{X}}}$ satisfy hermiticity conditions by forcing them to contain real and symmetric coefficients ($\underline{\underline{\mathbf{X}}} = \underline{\underline{\mathbf{X}}}^T$). In this way, they are decomposable into orthonormal eigenvectors, associated to real eigenvalues defining the anisotropy of each reactance tensor. It is also necessary to control the mean value of the reactance tensors, associated to the sum of its

diagonal elements and corresponding to the trace operator $\mathrm{Tr}(\underline{X}) = X_{sw}$. Following a holographic approach, a phase matching is finally realized by a modulation allowing the compensation of the phase of the reference wave and the creation of the objective one:

$$\mathbf{X}_{\rho\rho} = X_{sw}\left(1 + a_X^{\max} \mathbf{M}_{\rho\rho} \sin\left(\arg\left(\mathbf{E}_\rho/\mathbf{J}_\rho\right)\right)\right)$$
$$\mathbf{X}_{\rho\phi} = X_{sw} a_X^{\max} \mathbf{M}_{\rho\phi} \sin\left(\arg\left(\mathbf{E}_\phi/\mathbf{J}_\rho\right)\right)$$
$$\mathbf{X}_{\phi\rho} = \mathbf{X}_{\rho\phi} \tag{3}$$
$$\mathbf{X}_{\phi\phi} = X_{sw}\left(1 - a_X^{\max} \mathbf{M}_{\rho\rho} \sin\left(\arg\left(\mathbf{E}_\rho/\mathbf{J}_\rho\right)\right)\right).$$

The amplitude modulation of the generated hologram plays an important role in the conversion efficiency of the reference wave into a radiated wave[37,50]. The objective is thus to couple independently the orthogonal components of an incident wave along the preferred directions corresponding to the eigenvectors of the reactance tensor at each point of the metasurface. The anisotropy of the reactance tensors is finally defined by the constant $a_X^{\max}$, determined as a function of the values achievable by the synthesized patterns, as developed in the next section. A modulation of the anisotropy is performed in order to satisfy the polarization constraints of the reference and objective fields. The distributions $\mathbf{M}_{\rho\rho} \in [0,1]$ and $\mathbf{M}_{\rho\phi} \in [0,1]$ are thus determined by normalizing the magnitude of the following relations:

$$\mathbf{M}_{\rho\rho} = \left(\left|\frac{\mathbf{E}_\rho}{\mathbf{J}_\rho}\right| / \max\left(\left|\frac{\mathbf{E}}{\mathbf{J}}\right|\right)\right)^\alpha \tag{4}$$

$$\mathbf{M}_{\rho\phi} = \left(\left|\frac{\mathbf{E}_\phi}{\mathbf{J}_\rho}\right| / \max\left(\left|\frac{\mathbf{E}}{\mathbf{J}}\right|\right)\right)^\alpha \tag{5}$$

The $\alpha$ parameter (set at $\alpha = 0.5$ in this work) finally allows the adjustment of the spatial variation of anisotropy. By matching the decay of the surface wave with an increasing reactance modulation, the effective area occupied by the leaky wave can be optimized to increase the gain of the synthesized antennas[37,50].

## Anisotropic pattern synthesis

Following the definition of objective reactance tensors allowing the conversion of a reference wave into one or various radiation objectives, it is now necessary to propose a technique enabling the automated synthesis of patterns able to satisfy the desired electromagnetic constraints. A procedural generation technique is developed in this work leveraging the reaction-diffusion principle introduced by Alan Turing[1].

The principle is based on the interaction between antagonistic chemical species that he referred to as "morphogens", able to diffuse in space and interact to form biological patterns. Following such mechanisms, the Gray-Scott model[13] is harnessed here for its ease of implementation. This dynamic system is associated with the evolution of two populations of morphogens $U$ and $V$, converted into the following differential system:

$$\frac{\partial \mathbf{U}}{\partial t} = \mathbf{d_u}\nabla^2\mathbf{U} - \mathbf{U}\mathbf{V}^2 + \mathbf{f}(1 - \mathbf{U}) \tag{6}$$

$$\frac{\partial \mathbf{V}}{\partial t} = \mathbf{d_v}\nabla^2\mathbf{V} + \mathbf{U}\mathbf{V}^2 - (\mathbf{f} + \mathbf{k})\mathbf{V}. \tag{7}$$

The values $\mathbf{d_u}$ and $\mathbf{d_v}$ correspond to the diffusion coefficients associated with the Laplacian operator $\nabla^2$. $\mathbf{f}$ (feed) represent the spontaneous generation rate of morphogens $U$ and $\mathbf{k}$ (kill) corresponds to an extinction rate of the morphogens $V$. The common term $\mathbf{U}\mathbf{V}^2$ corresponds to the probability of conversion of a morphogen $U$ by

two $V$, thus subtracted from the concentrations $\mathbf{U}$ and added to the concentrations $\mathbf{V}$. Depending on the value of $\mathbf{f}$, the continuous emergence of morphogen $U$ is controlled by applying the generation rate to the term $(1 - \mathbf{U})$. Following a proper initialization, it is thus not possible to exceed a concentration of 1 because this term becomes less and less active as $\mathbf{U}$ increases. For the same purpose of morphogen population control, it is necessary that $V$ elements are removed faster than $U$ elements are created, making the total exctinction rate always greater than the generation rate via the term $-(\mathbf{f} + \mathbf{k})\mathbf{V}$. An explicit Euler scheme is implemented for the resolution of this system. Inspired by the ability of living organisms to locally self-organize in response to external stimuli or to fulfill biological functions, we are interested in exploiting these generative models to ensure the synthesis of structures with desired electromagnetic properties. Many combinations guarantee the generation of circular patterns whose reproduction resembles that of cells, occupying the available space following a succession of divisions. This type of Turing pattern is retained in this work, exploiting its ability to automatically form self-similar and compact arrangements that can be used to ensure local control of reactance on metasurfaces. Overall, this generation technique is also particularly well suited to meet the definition of boundary conditions prohibiting pattern growth.

This generative model thus provides an easy way to impose the types and dimensions of Turing patterns while respecting arbitrary boundary conditions. In connection with the isotropic diffusion of the morphogens computed with the Laplacian operator, the circular shapes of the selected patterns do not, however, offer independent control of incident orthogonal polarizations by anisotropy effect, being unable to adjust the orientation of the generated elements. In order to break these local symmetries and to impose privileged polarizations for the coupling of waves, it is thus necessary to deform these patterns to make them elliptical. The Gray-Scott model can thus be modified, generating patterns with the help of anisotropic diffusion[17–19]:

$$\frac{\partial \mathbf{U}}{\partial t} = \mathbf{d_u}\nabla \cdot (\underline{\mathbf{D}}\nabla\mathbf{U}) - \mathbf{U}\mathbf{V}^2 + \mathbf{f}(1 - \mathbf{U}) \tag{8}$$

$$\frac{\partial \mathbf{V}}{\partial t} = \mathbf{d_v}\nabla \cdot (\underline{\mathbf{D}}\nabla\mathbf{V}) + \mathbf{U}\mathbf{V}^2 - (\mathbf{f} + \mathbf{k})\mathbf{V}. \tag{9}$$

The diffusion is thus carried out by defining for each point of the space an associated tensor $\underline{\mathbf{D}}$, decomposing the Laplacian operator into two successive gradients:

$$\nabla \cdot (\underline{\mathbf{D}}\nabla\mathbf{M}) = \begin{bmatrix} \frac{\partial}{\partial x} & \frac{\partial}{\partial y} \end{bmatrix} \begin{bmatrix} D_{xx} & D_{xy} \\ D_{yx} & D_{yy} \end{bmatrix} \begin{bmatrix} \frac{\partial \mathbf{M}}{\partial x} \\ \frac{\partial \mathbf{M}}{\partial y} \end{bmatrix}.$$

A direct correspondence between the eigenvectors of the diffusion tensor and those of the generated reactance tensors allows the following diagonalization:

$$\underline{\mathbf{D}} = \underline{\mathbf{R}}_\phi \, \mathrm{diag}\left(\boldsymbol{\lambda}_1, \boldsymbol{\lambda}_2\right) \underline{\mathbf{R}}_\phi^T, \tag{10}$$

where $\boldsymbol{\lambda}_1$ and $\boldsymbol{\lambda}_2$ correspond to the eigenvalues of the diffusion tensors $\underline{\mathbf{D}}$ and where $\underline{\mathbf{R}}_\phi$ remain the same rotation matrices as previously defined, containing local eigenvectors. This complete generative model may seem complex at first glance but its implementation is remarkably straightforward using finite difference models. As developed in section 2 of the supplementary materials, a series of temporal iterations associated with the exploitation of discrete Laplacian operators enables the rapid emergence of complex patterns constrained in shape, size and orientation by the definition of local morphogenetic parameters. Following the definition of a generative model allowing the automated synthesis of anisotropic patterns, it is then

necessary to characterize the electromagnetic properties associated with the generated shapes.

## Characterization of the anisotropic patterns

In this work, elliptical patterns are created using the parameters $(f, k) = (0.032, 0.063)$. The diffusion constants $d_u = 0.95$ and $d_v = d_u/2$ are fixed. The patterns are generated on a $68 \times 68$ pixel grid with a supercell dimension of 13 mm, according to a preliminary study associated with the properties of the substrate presented below and the operating frequency of 20 GHz. These dimensions are chosen to obtain a desired average surface reactance, influencing the spatial frequencies of the printed distributions and the compatibility with chemical etching technologies. The patterns are designed with periodic boundary conditions to simulate infinite media without geometric breaks. This is achieved by applying a discrete Laplacian operator with a spatially periodic convolution product. Two degrees of freedom are utilized to modulate the properties of the reactance tensors. The largest eigenvalue of the diffusion tensor, $\lambda_1$, determines the anisotropy of the patterns. The trace of the diffusion tensor is normalized, imposing $\lambda_2 = 1 - \lambda_1$, thereby reducing the number of variables. A thresholding of the Turing patterns is performed to achieve a binary conversion and adjust the gaps between the elliptic shapes, introducing a second degree of freedom. After upscaling by a factor of 4 to limit staircase effects and normalizing the patterns, only the pixels satisfying $|\mathbf{V} - \sigma + 0.01| > \sigma$ are retained, where $\mathbf{V}$ represents the morphogens. Thresholding accelerates the characterization phase by limiting the number of generations to variations of $\lambda_1$. The addition of a supplementary threshold with a value of 0.01 avoids the appearance of a coalescence phenomenon between close patterns, especially when the threshold is at its lowest values. For the same reasons, we impose a limitation of the eigenvalues of the diffusion coefficients such as $\lambda_1 \in [0.15, 0.85]$, thresholds beyond which the generated elliptical patterns tend to locally merge under the action of a large diffusion coefficients. Additionally, the eigenvalues of the diffusion coefficients are limited to $\lambda_1 \in [0.15, 0.85]$ to prevent local merging of elliptical patterns caused by large diffusion coefficients. Following the generation of a pattern for a set of parameters, it is then possible to export the obtained geometry to the surface of a dielectric, forming metallic elliptical grains allowing the synthesis of reactance tensors by a homogenization effect. The simulations are performed on a Rogers RO3003 substrate with a relative permittivity $\epsilon_{r_1} = 3$, a loss tangent $\tan(\delta) = 10^{-3}$ and a thickness $h = 1.52$ mm. This substrate is chosen to facilitate the fabrication of proofs of concept, exploiting the relatively low permittivity to limit the dispersion of the anisotropy properties and the dimensions of the patterns to be etched. On the other hand, a low permittivity implies a reduced ability to modulate the anisotropy of the synthesized tensors. The exported patterns are finally be characterized by electromagnetic simulation using a finite element method in the frequency domain, implemented using the commercial software CST Studio Suite. By imposing periodic boundary conditions compatible with the constraints imposed during the generation of the patterns, an infinite medium is characterized by means of a Floquet port exciting two orthogonal plane waves in normal incidence. These simulations are used to extract 4 complex terms of scattering parameters for each simulated frequency, corresponding to the interaction between two orthogonal transverse electromagnetic excitations in emission and reception oriented according to the reference axes imposed by the simulation. The impedance reconstructed under these illumination conditions differs from that encountered by the dominant transverse magnetic excitation. A conversion method proposed by Patel and Grbic[51] is thus exploited to determine the reactance tensor directly useful for the conversion of surface waves into radiated waves. To limit the number of computations, these transformations are performed on the eigenvalues of these tensors

only, according to the eigendirections imposed by the diffusion tensors. Following a series of generations varying the parameters $\lambda_1$ and $\sigma$ previously introduced, it is finally possible to characterize the reactances $X_1$ and $X_2$ according to the main directions imposed for each pattern. To facilitate the interpretation and processing of these results, the reactance values obtained along the principal directions of the tensors are transformed into the average reactance seen by the surface wave $X_{sw}$ and the anisotropy modulation coefficient $a_X$, respectively determined as:

$$X_{sw} = \frac{X_1 + X_2}{2} \tag{11}$$

$$a_X = \frac{|X_1 - X_{sw}|}{X_{sw}} \tag{12}$$

To ensure the local generation of the desired electromagnetic characteristics on a metasurface, it is finally necessary to determine the parameters $(\lambda_1, \sigma)$ generating each couple of values $(X_{sw}, a_X)$. The collection of all these results finally allowed to determine a polynomial fitting revealing the parameters ensuring the synthesis of patterns associated to electromagnetic tensors with an average of $X_{sw} = 270\,\Omega$ and a maximum reactance modulation $a_X^{max} = 0.186$. These data and the details of the fitting process are developed in section 3 of the supplementary materials. The synthesis of the metasurfaces is finally ensured by determining the local morphogenetic parameters associated with the desired electromagnetic constraints, guiding the growth of anisotropic Turing patterns obtained by solving by finite differences the model presented in Eqs. (8) and (9).

## Measurement Techniques

Realized gain measurements are performed around 20 GHz in an anechoic chamber. Far-field conditions are obtained using a parabolic reflector and allow the measurement of two orthogonal polarizations. Numerical processing is then applied to convert this data into circular polarizations. The uncertainty of gain measurement is evaluated at $\pm 0.6$ dB. The near-field measurements are performed using a Cartesian scanner equipped with an open-ended waveguide designed for the K band (18–26.5 GHz). A scan is performed in a plane parallel to the radiating aperture at a distance of 20 cm from the latter, according to an identical stroke for each axis of a total length of 275 mm sampled every 5 mm (or 51 points per axis). The field is measured successively according to the vertical and horizontal polarizations by rotating the waveguide, allowing the data to be represented with circular polarizations by linear combinations of the measurements.

## Data availability

The data required to reproduce the results presented in this paper are detailed in sections 4, 5, and 6 of the supplementary materials accompanying this paper. All other data that support the plots within this paper and other findings of this study are available from the corresponding author upon request.

## Code availability

The code that support the findings of this study are available from the corresponding author upon request. A detailed analysis of the models developed in this paper is given in sections 1, 2, and 3 of the supplementary materials.

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

## Acknowledgements
The work of Thomas Fromenteze was supported by the ANR JCJC MetaMorph ANR-21-CE42-0005. The work of Okan Yurduseven was supported by the Leverhulme Trust under Research Leadership Award RL-2019-019. Experiments were performed at the PLATINOM platform of the University of Limoges, France, supported by the European regional development foundation and the Nouvelle-Aquitaine region (FEDER-PILIM 2015-2020 Nouvelle-Aquitaine). We thank Marc Thevenot for his advice on electromagnetic simulations.

## Author contributions
T.F. designed the generative model with initial inputs from D.S. and periodic discussions with O.Y. and C.D. T.F. performed the characterization of the generated properties and the synthesis of the metasurfaces. C.H. and T.F. carried out a comparative study with conventional techniques. E.A. and T.F. carried out the experiments and post-processing of data. T.F. wrote a first draft, then revised with all the authors.

## Competing interests
The authors declare no competing interests.
