## [Peer Review File · Nature Communications]

Morphogenetic Metasurfaces Unlocking the Potential of Turing PatternsREVIEWER COMMENTS

Reviewer #1 (Remarks to the Author):

In this manuscript, the authors proposed an interesting way to design metasurfaces based on the reaction-diffusion principle. Such a principle can be traced back to the 1950s when Alan Turing attempted to explain the structuring of living organisms (i.e., morphogenesis). The RF metasurfaces described in the manuscript can radiate electromagnetic signals into the free space by spatially tailoring the imaginary part of the surface impedance. This is achieved by applying weak perturbations to the guided surface waves via nanopatterning the top surface of the waveguide. To generate the proposed morphogenetic metasurfaces, the authors first build the connection of a set of morphogenetic parameters to the electromagnetic parameters of the generated periodic pattern with the help of FEM simulations. Second, the authors convert a specific metasurface function (e.g., beam steering, holography, etc.) into local electromagnetic parameter distributions (i.e., the so-called objective reactance tensors). Finally, the authors further convert the determined objective reactance tensors into spatially variant morphogenetic parameters to generate the metasurface pattern. Such a design principle is then verified by creating functional metasurfaces for polarization-dependent beam steering as well as near-field holography. The reported experimental results are solid and convincing.

However, at least for the current manuscript version, my major concern is that the authors didn't fully clarify the (potential) value of this proposed method to a level that unambiguously justifies its publication in Nature Communications. In other words, the authors should demonstrate a case (could even be a very special one) where the morphogenetic metasurfaces can perform better than the metasurfaces designed in a conventional way. In my personal opinion, the demonstrated far-field beam steering as well as near-field holography are very typical metasurface applications that can be designed using the conventional method without any problem. In fact, after reading the current manuscript, I didn't see a clear advantage of this morphogenetic method beyond relaxing the rigorous periodic condition used in the conventional method. If that is the case, the authors should then try to find a specific scenario where periodic metasurfaces are highly unfavorable. If that is not the case, the authors should point out the other, more important

advantages more clearly.

I also have some minor suggestions that the authors may consider as follows:

1) The figure captions for Fig. 1 and Fig. 2 are too simple. They don't provide enough information to allow readers to understand the material shown in the figure. The authors should consider adding more detailed descriptions.

2) In page 3, the authors write "The populations of U and V evolve according to the spontaneous generation coefficient f (feed) for the V morphogens and spontaneous extinction k (kill) for the U morphogens." I believe the generation coefficient (f) should be for U and the extinction coefficient (k) should be for V. The authors should double-check this.

3) I wonder if the morphogenetic method can mitigate the cross-talk issue found in metasurfaces designed by the conventional method (e.g., <https://doi.org/10.1002/lpor.202000448>). This can be a real advantage if that is the case.

4) I am confused why the authors claim "The exploitation of morphogenesis-inspired generative models proves particularly well suited for solving inverse design problems". I could not find an inverse design problem solved in the manuscript. In fact, by converting the target metasurface function into certain objective reactance tensors, this is already a forward-direction design...

Reviewer #2 (Remarks to the Author):

All the fascinating functions of metasurfaces are finally attributed to their artificial patterns. In this study, for the first time the authors bring the reaction-diffusion model into the synthesis of metasurfaces, which makes it possible for automatic meta-atom generation without gradient descent optimization to meet the specific near/far field radiation characteristics.

The study first translates the wavefront shaping function into the surface reactance tensor

of each meta-atom. Then the reactance tensor is transferred to the diffusion tensor in the Gray-Scott reaction-diffusion model to guide anisotropic pattern generation. Finally, the correspondence between the generated patterns and their electromagnetic responses is built by a series of electromagnetic simulations, and proper atoms are gathered into the final metasurface.

Such morphogenetic metasurfaces are of obvious innovation from the following aspects. The generated metasurface pattern is very different from the usual ones, which may offer richer electromagnetic responses. The proposed model can handle isotropic and also anisotropic radiation by introducing the diffusion tensor. The examples of beam deflection and holographic imaging prove the soundness of the methodology, which is a good contribution to the metasurface field. The manuscript can be accepted provided the following issues are addressed.

1. Each pattern is generated with periodic boundary consideration. When different patterns are stitched together, what is the impact of their coupling and the impact of discontinuous patterns?
2. As compared to traditional metasurfaces made of simple geometric shapes, the morphogenetic metasurfaces show smaller feature size with each meta-atom containing many ellipses. Can this design be scaled to optical frequencies considering the fabrication limit? What is the applicable frequency range of this concept.
3. It seems one still needs to build a library to map the reactance to the pattern through tedious electromagnetic simulations in Fig. S10. From this point of view, the design complexity is similar to traditional one. It is not clear why this method is well suited for solving inverse design problems as the authors mentioned in the beginning.
4. The study here is concentrated to surface wave to leaky wave coupling, which can be done by surface reactance engineering. Is this method general enough to deal with wavefront shaping by transmission or reflection through either metallic or dielectric metasurfaces? In those cases, the modulation is usually done to the Jones matrix of the meta-atoms. Is that possible to relate the Jones matrix to the diffusion tensor and the Gray-Scott model?

1 Reviewer 1

In this manuscript, the authors proposed an interesting way to design metasurfaces based on the reaction-diffusion principle. Such a principle can be traced back to the 1950s when Alan Turing attempted to explain the structuring of living organisms (i.e., morphogenesis). The RF metasurfaces described in the manuscript can radiate electromagnetic signals into the free space by spatially tailoring the imaginary part of the surface impedance. This is achieved by applying weak perturbations to the guided surface waves via nanopatterning the top surface of the waveguide. To generate the proposed morphogenetic metasurfaces, the authors first build the connection of a set of morphogenetic parameters to the electromagnetic parameters of the generated periodic pattern with the help of FEM simulations. Second, the authors convert a specific metasurface function (e.g., beam steering, holography, etc.) into local electromagnetic parameter distributions (i.e., the so-called objective reactance tensors). Finally, the authors further convert the determined objective reactance tensors into spatially variant morphogenetic parameters to generate the metasurface pattern. Such a design principle is then verified by creating functional metasurfaces for polarization-dependent beam steering as well as near-field holography. The reported experimental results are solid and convincing.

However, at least for the current manuscript version, my major concern is that the authors didn't fully clarify the (potential) value of this proposed method to a level that unambiguously justifies its publication in Nature Communications. In other words, the authors should demonstrate a case (could even be a very special one) where the morphogenetic metasurfaces can perform better than the metasurfaces designed in a conventional way. In my personal opinion, the demonstrated far-field beam steering as well as near-field holography are very typical metasurface applications that can be designed using the conventional method without any problem. In fact, after reading the current manuscript, I didn't see a clear advantage of this morphogenetic method beyond relaxing the rigorous periodic condition used in the conventional method. If that is the case, the authors should then try to find a specific scenario where periodic metasurfaces are highly unfavorable. If that is not the case, the authors should point out the other, more important advantages more clearly.

We would like to thank you for the relevance of this review. In the initial version of the manuscript, the positioning of this approach in relation to conventional techniques developed in the literature is indeed not presented. As explained above, this work has been initiated more recently so that future publications related to these techniques can quantify the advantages linked to the self-structuring of the patterns generated, as well as to the introduction of partial disorder.

Constraints on the arrangement of meta-atoms are notoriously limiting for the synthe-

sis of anisotropic metasurfaces. Elliptical shapes offer many advantages, but the spacings formed between them when arranged according to uniform grids become highly dependent on the local orientations imposed. In line with the work led by S. Torquato in condensed matter theory [1, 2, 3], the compact arrangement of elliptical patterns tends to be optimized by the introduction of partial disorder [4], for which generative methods inspired by morphogenesis are particularly well suited. We have recently published an article on this subject linked to the formation of isotropic electromagnetic bandgaps:

Chehami, F., Decroze, C., Pasquet, T., Perrin, E., and Fromenteze, T., Morphogenetic Design of Self-Organized Correlated Disordered Electromagnetic Media. ACS Photonics, Publication Date: May 24, 2023

By proposing a method for generating self-structured patterns to ensure more regular spacing, while satisfying dimensional and orientation constraints, we create more favorable conditions for homogenizing the properties obtained. This point is also directly related to your remark about the break in pattern periodicity.

In order to provide first elements able to convince readers of the advantages of this approach in this first paper, the following comment has been added to the Discussion section:

The spatial self-structuring properties of the proposed technique enable more densely and uniformly arranged spatial distributions to be generated, compared with conventional methods based on periodic arrangements of meta-atoms. Some advantages linked to these properties are presented in the supplementary materials, but in-depth studies will be needed to fully quantify the benefits of this new technique.

An additional section "Positioning in relation to conventional techniques" has been added to the Supplementary materials and copied at the end of this response.

I also have some minor suggestions that the authors may consider as follows:

1) The figure captions for Fig. 1 and Fig. 2 are too simple. They don't provide enough information to allow readers to understand the material shown in the figure. The authors should consider adding more detailed descriptions.

Following your recommendation, we propose two more complete descriptions to make these figures self-sufficient.

- The initial caption of Fig. 1:

Figure 1: Interaction modeling of morphogens capable of reaction (**A**) and diffusion (**B**)

for the formation of Turing patterns (**C**) by means of a differential system (**D**). The introduction of diffusion tensors (**E**) paves the way to the procedural synthesis of anisotropic media (**F**).

is replaced by the following:

Figure 1: The Gray-Scott model is based on the simulation of virtual, antagonistic chemical compounds referred to by A. Turing as "morphogens". Following simple reaction (**A**) and diffusion (**B**) mechanisms, these morphogens, separated into two categories U and V , form a system of prey and predator, providing the opportunity, by adjusting certain parameters, to establish population equilibria generating spatial patterns, known as Turing patterns (**C**). The stochastic rules defined for each particle lead to a set of two partial differential equations on a macroscopic scale, shifting from a Lagrangian description of the process to an Eulerian perspective defining a concentration field of morphogens **U** and **V** (**D**). The diffusion constants d_u and d_v control the dimensions of the patterns generated, while the pair of parameters f and k influence their type. This model is modified to offer a new degree of freedom with anisotropic characteristics. Each point in space is thus associated with a diffusion tensor (**E**) whose eigenaxes determine preferred directions influencing the local orientation of the synthesized patterns (**F**).

- The initial caption of Fig. 2:

Figure 2: (**Top**) Study of the conversion of morphogenetic parameters into synthesized electromagnetic properties. (**Bottom**) Generation of a morphogenetic metasurface by adapting the electromagnetic characteristics defined by a radiation objective to the parameters ensuring the growth of anisotropic Turing patterns. The metasurface synthesis is represented for the different quarters as the time iterations progress.

is replaced by the following:

Figure 2: The synthesis of a morphogenetic metasurface is carried out in two stages. A first characterization phase (**Top**) enables generation parameters to be associated with the electromagnetic properties synthesized. The study is restricted to the growth of cellular patterns whose ellipticity, dimensions and orientation are controlled by local constraints influencing morphogen self-structuring. The parameters are identical in space, which is defined with periodic boundary conditions to obtain spatial distributions enabling continuous tessellation of the plane. Finite element simulation is used to extract the electromagnetic characteristics associated with each set of morphogenetic parameters. Numerical processing is then applied to derive the reactance tensors required to convert surface waves into radiated waves. In a second phase (**Bottom**), the definition of radiation objectives enables the desired electromagnetic characteristics to be derived at the

surface of a parallel plate waveguide excited by a monopole. Conversion of these targets into local morphogenetic parameters is required for metasurface generation. An iterative resolution of the anisotropic Gray-Scott model ensures the growth of elliptical patterns undergoing a series of successive divisions until they occupy the entire permitted space. Under the influence of the morphogenetic parameters, the elements self-structure to synthesize the desired electromagnetic tensors.

2) In page 3, the authors write “The populations of U and V evolve according to the spontaneous generation coefficient f (feed) for the V morphogens and spontaneous extinction k (kill) for the U morphogens.” I believe the generation coefficient (f) should be for U and the extinction coefficient (k) should be for V . The authors should double-check this.

We thank you for spotting this error in contradiction with the information presented in Fig. 1 and with the requirements necessary to obtain population equilibria responsible for pattern formation. This typo has been corrected in the new version of the manuscript.

3) I wonder if the morphogenetic method can mitigate the cross-talk issue found in metasurfaces designed by the conventional method (e.g., <https://doi.org/10.1002/lpor.202000448>). This can be a real advantage if that is the case.

The cross-talk problem described in this work does indeed seem to be linked to some of the properties provided by this generative model. Further investigations will be needed to fully quantify these effects. Some of the points discussed concerning the advantage of the morphogenetic approach over conventional methods appear to be at least partly related to this issue.

The proposed generation method ensures the compact and regular arrangement of varying elliptical shapes. Non-intuitively, the introduction of a certain degree of disorder within a periodic arrangement facilitates the homogenization of the properties obtained, resulting in gaps that are much more regular despite the rotation of the patterns, compared with conventional techniques based on uniform arrangements. In this sense, cross-talk is more easily controlled, offering for each meta-atom of a metasurface interactions with its neighbors much closer to those obtained during the characterization phase considering periodic boundary conditions and quasi-identical atoms.

The following sentence has been added to the Discussion section of the main paper:

Although these demonstrations are presented in the microwave range, a direct transposition of the proposed techniques is of course conceivable in the photonic domain, adapting the reference wave and radiation objectives as required. The self-structuring capability of meta-atoms made of dielectric materials would offer a potential answer to the limitations

associated with periodicity and geometric breaks in uniform lattices, enabling the fundamental limits of conventional metasurfaces to be re-explored [5].

4) *I am confused why the authors claim “The exploitation of morphogenesis-inspired generative models proves particularly well suited for solving inverse design problems”. I could not find an inverse design problem solved in the manuscript. In fact, by converting the target metasurface function into certain objective reactance tensors, this is already a forward-direction design...*

This point was also raised by the second reviewer, prompting us to reconsider this claim. Our initial reasoning was based on the fact that, unlike conventional approaches, the proposed method does not directly convert electromagnetic properties into geometric parameters via a lookup table, but tends to impose local constraints guiding the generation of elements capable of interacting with each other to find the best possible arrangement. The proposed method is not entirely a forward design technique in the sense that some form of collective optimization is performed to satisfy the desired electromagnetic constraints.

To limit confusion and better describe the contributions of the proposed technique, the positioning relative to inverse design has been abandoned in favor of the principle of generative design, which seems more appropriate. Future investigations will enable us to better position this work in relation to inverse design techniques, in order to clear up these ambiguities.

Thus, the following sentence in the abstract:

”The exploitation of morphogenesis-inspired generative models proves particularly well suited for solving inverse design problems, converting global physical constraints into local interactions of simulated chemical reactants ensuring the emergence of self-organizing meta-atoms.”

is modified as follows:

”The exploitation of morphogenesis-inspired models proves particularly well suited for solving generative design problems, converting global physical constraints into local interactions of simulated chemical reactants ensuring the emergence of self-organizing meta-atoms.”

The references used to position the work in relation to inverse design are also replaced by applications oriented towards shape optimization and generative design guided by physical constraints and the following comments related to the limitations of inverse

design techniques have also been removed.

We thank you for the time you have devoted to analyzing this work, and hope that the information we have provided has met your expectations.

2 Reviewer 2

All the fascinating functions of metasurfaces are finally attributed to their artificial patterns. In this study, for the first time the authors bring the reaction-diffusion model into the synthesis of metasurfaces, which makes it possible for automatic meta-atom generation without gradient descent optimization to meet the specific near/far field radiation characteristics. The study first translates the wavefront shaping function into the surface reactance tensor of each meta-atom. Then the reactance tensor is transferred to the diffusion tensor in the Gray-Scott reaction-diffusion model to guide anisotropic pattern generation. Finally, the correspondence between the generated patterns and their electromagnetic responses is build by a series of electromagnetic simulations, and proper atoms are gathered into the final metasurface.

Such morphogenetic metasurfaces are of obvious innovation from the following aspects. The generated metasurface pattern is very different from the usual ones, which may offer richer electromagnetic responses. The proposed model can handle isotropic and also anisotropic radiation by introducing the diffusion tensor. The examples of beam deflection and holographic imaging prove the soundness of the methodology, which is a good contribution to the metasurface field. The manuscript can be accepted provided the following issues are addressed.

We would like to thank you for this comprehensive analysis of our work and for the relevant comments. Responses are provided below each remark, specifying which parts of the main article or supplementary material have been modified where appropriate.

1. Each pattern is generated with periodic boundary consideration. When different patterns are stitched together, what is the impact of their coupling and the impact of discontinuous patterns?

Under the effect of varying morphogenetic parameter fields during the synthesis of a metasurface, patterns undergo rotations and migrations, requiring each element to be surrounded by neighbors with different geometric characteristics, in contrast to characterization conditions where shapes are almost invariant and arranged as best they can to pave the plane. The new positioning proposed, comparing the performance of a simplified morphogenetic metasurface with one generated in a more conventional way, highlights the beneficial effect of smaller geometry discontinuities. These differences can in fact be observed as early as the characterization stage, studying the effect of pattern rotation on the variation of the synthesized impedance tensors. The approach developed thus ensures a more regular arrangement of the generated ellipses when the patterns are rotated.

At the scale of full metasurfaces, where reactance tensor fields vary spatially impos-

ing geometric discontinuities associated with pattern rotation and migration, the effect on synthesized parameters is trickier to characterize in situ. Comparison of the radiation from a morphogenetic metasurface and a conventional equivalent shows once again, however, that polarization control is more advanced, as is reference wave conversion for patterns assumed to have the same average performance. While the coupling effect between different meta-atoms has an impact in each case, the much more uniform variations in the morphogenetic case, as well as the better performance observed, highlight the improved control of the electromagnetic properties synthesized.

Following your recommendations, and those of the second reviewer, we have added a section to the supplementary materials offering some simple comparisons of the effects of these geometry breaks. This section is also copied at the end of this reply.

The following comment has been added to the Discussion section:

The spatial self-structuring properties of the proposed technique enable more densely and uniformly arranged spatial distributions to be generated, compared with conventional methods based on periodic arrangements of meta-atoms. Some advantages linked to these properties are presented in the supplementary materials, but in-depth studies will be needed to fully quantify the benefits of this new technique.

2. As compared to traditional metasurfaces made of simple geometric shapes, the morphogenetic metasurfaces show smaller feature size with each meta-atom containing many ellipses. Can this design be scaled to optical frequencies considering the fabrication limit? What is the applicable frequency range of this concept.

Based on our knowledge of photonic fabrication technologies, the proposed approach does not appear to impose any additional restrictions when compared with the design of more conventional metasurfaces, but is accompanied by properties that could, on the contrary, facilitate such fabrications. As illustrated in the comparison provided with this response and in the new version of the manuscript, synthesized patterns with comparable performance have dimensions of the same order of magnitude and provide a better control of the inter-element spacings. Moreover, the introduction of partial disorder into the arrangement of meta-atoms seems to offer greater robustness to manufacturing uncertainties [6], being less sensitive to the perturbation of symmetry axes absent from these designs.

We recently published a paper using an isotropic reaction-diffusion model to synthesize materials with controlled level of disorder for the generation of isotropic band gaps:

Chehami, F., Decroze, C., Pasquet, T., Perrin, E., and Fromenteze, T., Morphogenetic

It therefore seems possible to directly transpose the proposed generation techniques to nanophotonic technologies, considering the necessary adaptation of meta-atoms to dielectric materials more suitable for these frequency bands in view of the increase in metal losses. If it is easy to generate surface waves on microwave systems, converting a reference wave radiated from a source external to the substrate is also a valid and compatible alternative, thus simplifying electromagnetic characterization to the excitation of incident plane waves alone.

In connection with these comments, we have added the following to the discussion section of the paper:

Recent work has shown that these techniques can be used to synthesize partially correlated disordered media [7]. Eliminating the need for axes of symmetry by introducing controlled disorder would, via the proposed morphogenetic technique, provide a means of conceiving designs that could be more robust to fabrication errors [6].

3. It seems one still need to build a library to map the reactance to the pattern through tedious electromagnetic simulations in Fig. S10. From this point of view, the design complexity is similar to traditional one. It is not clear why this method is well suited for solving inverse design problems as the authors mentioned in the beginning.

This point was also raised by the first reviewer, prompting us to reconsider this claim. Our initial reasoning was based on the fact that, unlike conventional approaches, the proposed method does not simply convert electromagnetic parameters into geometric parameters via a lookup table, but tends to impose local constraints guiding the generation of elements capable of interacting with each other to find the best possible arrangement. The proposed method is not entirely a forward design technique in the sense that the procedure is not applied independently for each element, but forms a collective optimization of self-arranging patterns to satisfy the desired electromagnetic constraints.

In order to limit confusion and better describe the benefits of the proposed technique, we have abandoned the term "inverse design" in favor of "generative design", which seems more appropriate. Future investigations will enable us to better position this work in relation to inverse design techniques, in order to clear up these ambiguities.

Thus, the following sentence in the abstract:

"The exploitation of morphogenesis-inspired generative models proves particularly well suited for solving inverse design problems, converting global physical constraints into lo-

cal interactions of simulated chemical reactants ensuring the emergence of self-organizing meta-atoms.”

is modified as follows:

”The exploitation of morphogenesis-inspired models proves particularly well suited for solving generative design problems, converting global physical constraints into local interactions of simulated chemical reactants ensuring the emergence of self-organizing meta-atoms.”

The references used to position work related to inverse design are also replaced by applications oriented towards generative design guided by physical constraints.

4. The study here is concentrated to surface wave to leaky wave coupling, which can be done by surface reactance engineering. Is this method general enough to deal with wavefront shaping by transmission or reflection through either metallic or dielectric metasurfaces? In those cases, the modulation is usually done to the Jones matrix of the meta-atoms. Is that possible to relate the Jones matrix to the diffusion tensor and the Gray-Scott model?

This remark appears perfectly valid to us, and is directly related to current investigations. There are no restrictions linked to the generative method imposing the sole conversion of surface waves, being able to synthesize objective tensors resulting from interaction with arbitrary reference waves potentially originating from a source outside the substrate, whether the meta-atoms have been characterized in transmission or reflection. As explained above, this characterization step can even be simplified by considering that it is no longer necessary to transform characterization by an incident plane wave from a Floquet port into interaction with a magnetic transverse surface wave. Jones matrices find a direct correspondence with impedance tensors in this case, enabling the diffusion tensors guiding metasurface synthesis to be adapted by extracting eigenvalues and eigenvectors for each point in space.

We have added a comment on the link between reactance tensors and Jones matrices in the supplementary materials, as well as an interesting reference on the subject:

A direct analogy exists with the Jones matrices defined for metasurfaces used in transmission, allowing advanced control of polarization conversion [8].

We’d like to thank you for the time you’ve invested in this review, and hope that we’ve been able to answer all your questions.

Sincerely,
Thomas Fromentèze

The following section has been added to the supplementary materials to provide elements of comparison between the properties of conventional metasurfaces based on uniform arrangements of meta-atoms and the proposed approach:

Positioning in relation to conventional techniques

Numerous works based on the exploitation of patterns arranged on uniform grids already exist in the literature. The most advanced proofs of concept exploit sophisticated mathematical formalisms, notably allowing the effect of impedance modulations on the surface wave to be taken into account in order to correct the objective tensors. These optimizations are obtained at the cost of the development of complex numerical techniques and the use of restrictions on metasurface geometries to facilitate domain decompositions, but have proved their effectiveness and define the state-of-the-art in metasurface antennas.

In this section, it is proposed to compare the performance of the morphogenetic generation technique with a conventional approach, consisting in exploiting patterns arranged on a uniform grid. In order to offer a comparison that is both fair and easy to comprehend, it has been necessary to simplify the constraints considered. It is therefore proposed to compare the performance obtained for two metasurfaces occupying a maximum area of $75 \times 75 \text{mm}^2$, radiating a single left-hand circularly polarized beam in a direction normal to the metasurfaces. The first-order approximations previously developed are once again used to facilitate these comparisons.

For this simplified comparison, the anisotropy of the objective tensors is kept constant. All the elliptical patterns generated for the conventional metasurface will therefore be of identical dimensions, and the morphogenetic parameters will be kept identical throughout the metasurface. In both cases, only rotations of the patterns will be allowed, either directly or via the diffusion tensors to ensure surface wave conversion.

It is therefore important to note that the performance obtained in each case can be greatly optimized. Under these simplified conditions, it will nevertheless be possible to

easily illustrate the effect of geometric discontinuities linked to the exploitation of uniform grids of patterns, where the self-structuring capacity of morphogenetic patterns tends to provide a more homogeneous distribution of these spacings.

Firstly, it is proposed to compare the characteristics obtained for two patterns optimized to present comparable properties. The dimensions are chosen so as to ensure the synthesis of a reactance tensor whose average tends towards $X_{sw} = 270\Omega$ and whose anisotropy modulation tends to $a_X = 0.17$. The latter value is deliberately chosen to be lower than the maximum achievable by the morphogenetic method, in order to facilitate the synthesis of an equivalent characteristic by means of identical patterns arranged on a uniform grid. The first part of this study focuses on the effect of rotating the tensors to be synthesized under these simplified conditions. An illustration of the patterns generated in each case already reveals some significant differences (Fig. C1)

Figure C1: Patterns enabling the synthesis of reactance tensors of identical average and anisotropy for two rotations such as $\varphi = 0^\circ$ and $\varphi = 45^\circ$. The self-structuring capability of the morphogenetic technique results in a more regular distribution of elements.

For a rotation series varying between $\varphi = 0^\circ$ and $\varphi = 90^\circ$, the tensors are determined following the procedure presented previously. The average values X_{sw} of the reactance tensors obtained for each rotation value, together with the a_X anisotropy, are presented

Figure C2: Effect of geometric discontinuities on the variation in the characteristics of the reactance tensors generated.

in Fig. C2.

Under the effect of variations in distances between elements, the conventional approach undergoes a change in the synthesized characteristics. In line with the previous graphical justification, the properties obtained by the morphogenetic technique are more rotationally stable, guaranteeing better control of the properties obtained.

It is important to note that a homogenization technique was used to extract morphogenetic parameters. As the synthesis was carried out in a domain with periodic boundary conditions, initial results tended to indicate a strong dependence of pattern rotation on extracted properties. Due to the repulsion of elements on shared boundaries, the compactness of the resulting arrangements is influenced by the orientation of the eigenaxes. To limit these boundary condition effects and extract only the features associated with the patterns, the characterization space was increased to $20 \times 20\text{mm}^2$. Furthermore, the results presented are obtained by averaging 10 random initializations. Finally, these results are averaged for two orientations of the Floquet ports (following a 45° rotation), considering that results from a tensor diagonalization are assumed to be invariant with the rotation of the eigenaxes chosen for excitation. This property was verified for elliptical patterns arranged on a uniform grid, where rotation of the eigenaxes of the Floquet ports had no effect on the properties extracted after tensor diagonalization.

Two metasurfaces are synthesized from these patterns according to the specifications previously outlined (Fig. C3). These representations highlight the smaller dimensions of the patterns generated by reaction-diffusion compared with those obtained for uniform paving, recalling that the design was carried out to obtain the same average reactance

and anisotropy characteristics. This difference in dimensions is justified by the stronger interaction of the Turing patterns with neighboring elements. It should be noted, however, that the smallest spacings between these patterns remain generally greater than those obtained for the conventional approach when the ellipses are aligned horizontally, thus facilitating the fabrication of morphogenetic metasurfaces.

Figure C3: Conventional (left) and morphogenetic (right) metasurfaces designed according to the same specifications to radiate a beam normal to the antennas, circularly polarized to the left. The patterns are chosen in each case to provide the closest possible average characteristics.

The computation of radiation at 20 GHz highlights the effect of better control of the synthesized reactance tensors (Fig. C4). With these two basic designs, radiation is well achieved in the desired direction and polarization state. However, the morphogenetic method guarantees better control of polarization, both in terms of the level of secondary lobes in left circular polarization and overall control of cross-polarization.

Figure C4: Improved anisotropy control through the morphogenetic approach enables more selective management of the radiated polarization state. The color bar maximum is adjusted according to the maximum gain obtained for the pair of metasurfaces, ensuring that the colors correspond to identical gain levels.

With a more homogeneous distribution of the generated patterns, a slight increase in on-axis gain is finally obtained with this demonstration, as well as a widening of the operating band in connection with the introduction of periodicity breaks compared to the conventional approach (Fig. C5).

Figure C5: Under the simple conditions proposed, the morphogenetic approach enables an increase in gain and a widening of the bandwidth compared with the conventional approach.

Considering the simplicity of this demonstration, both approaches can obviously be greatly optimized, but a comparison carried out under simple, controlled conditions nevertheless highlights the contributions of the proposed technique. The multiplexing of radiation targets, the optimization of aperture efficiency and the synthesis of holograms tend to impose much stronger constraints on the spatial modulation of anisotropy, implying even greater constraints on the arrangement of the resulting meta-atoms (Fig. C6). Further investigations will enable us to quantify in finer detail the advantages offered by the proposed technique over conventional approaches in these more advanced cases.

Figure C6: Zoom in on a distribution of meta-atoms, highlighting the self-structuring capacity of the proposed morphogenetic technique.

References

- [1] O. U. Uche, S. Torquato, and F. H. Stillinger, “Collective coordinate control of density distributions,” *Physical Review E*, vol. 74, no. 3, p. 031104, 2006.
- [2] S. Torquato and F. H. Stillinger, “Local density fluctuations, hyperuniformity, and order metrics,” *Physical Review E*, vol. 68, no. 4, p. 041113, 2003.
- [3] Y. Jiao, T. Lau, H. Hatzikirou, M. Meyer-Hermann, J. C. Corbo, and S. Torquato, “Avian photoreceptor patterns represent a disordered hyperuniform solution to a multiscale packing problem,” *Physical Review E*, vol. 89, no. 2, p. 022721, 2014.
- [4] A. Donev, I. Cisse, D. Sachs, E. A. Variano, F. H. Stillinger, R. Connelly, S. Torquato, and P. M. Chaikin, “Improving the density of jammed disordered packings using ellipsoids,” *Science*, vol. 303, no. 5660, pp. 990–993, 2004.
- [5] C. Gigli, Q. Li, P. Chavel, G. Leo, M. L. Brongersma, and P. Lalanne, “Fundamental limitations of Huygens’ metasurfaces for optical beam shaping,” *Laser & Photonics Reviews*, vol. 15, no. 8, p. 2000448, 2021.

- [6] P. M. Piechulla, B. Fuhrmann, E. Slivina, C. Rockstuhl, R. B. Wehrspohn, and A. N. Sprafke, “Tailored light scattering through hyperuniform disorder in self-organized arrays of high-index nanodisks,” *Advanced Optical Materials*, vol. 9, no. 17, p. 2100186, 2021.
- [7] F. Chehami, C. Decroze, T. Pasquet, E. Perrin, and T. Fromenteze, “Morphogenetic design of self-organized correlated disordered electromagnetic media,” *ACS Photonics*, 2023.
- [8] N. A. Rubin, A. Zaidi, A. H. Dorrah, Z. Shi, and F. Capasso, “Jones matrix holography with metasurfaces,” *Science Advances*, vol. 7, no. 33, p. eabg7488, 2021.

REVIEWERS' COMMENTS

Reviewer #1 (Remarks to the Author):

I appreciate the authors for their great efforts in addressing my questions and concerns about this manuscript. They have excellently answered the questions to which they know the exact answer. They added additional comments and information in the revised manuscript to allow the readers to understand the material easier and position the paper in the field more clearly. Finally, I am happy to see that they are honest and objective in terms of sharing their opinions on things that are not clear or still debatable.

Overall, I am satisfied with the revised manuscript, and I can now recommend the publication of the paper.

Reviewer #2 (Remarks to the Author):

The authors have addressed most of my concerns in the responses. They have conducted additional comparison between the conventional periodic patterns and the self-organized patterns. The later designs are more uniform, more rotationally stable, and more robust to fabrication errors. When designed as a specific antenna, the self-organized pattern shows slightly increased gain and wider bandwidth. My only concern is that these minor improvements do not reflect the significant advances of this approach as compared to the conventional ones. Other examples should be found to exactly highlight the benefit of this method.

1 Reviewer 1

I appreciate the authors for their great efforts in addressing my questions and concerns about this manuscript. They have excellently answered the questions to which they know the exact answer. They added additional comments and information in the revised manuscript to allow the readers to understand the material easier and position the paper in the field more clearly. Finally, I am happy to see that they are honest and objective in terms of sharing their opinions on things that are not clear or still debatable.

Overall, I am satisfied with the revised manuscript, and I can now recommend the publication of the paper.

We would like to thank you sincerely for the time you have invested in analyzing our work, and for the pertinent comments that have helped to improve it.

2 Reviewer 2

The authors have addressed most of my concerns in the responses. They have conducted additional comparison between the conventional periodic patterns and the self-organized patterns. The later designs are more uniform, more rotationally stable, and more robust to fabrication errors. When designed as a specific antenna, the self-organized pattern shows slightly increased gain and wider bandwidth. My only concern is that these minor improvements do not reflect the significant advances of this approach as compared to the conventional ones. Other examples should be found to exactly highlight the benefit of this method.

In line with your last remark, we have carried out a comparison of a more significant case by increasing the dimensions of the generated metasurfaces. This case, closer to a realistic application, provides data demonstrating the advantage of the proposed technique for controlling the synthesized electromagnetic characteristics. In line with the previous comparison, maximum gain, frequency control and bandwidth are improved thanks to the restricted effect of pattern periodicity breaks.

The updated results are shown below:

Figure C1: Conventional (left) and morphogenetic (right) metasurfaces designed according to the same specifications to radiate a beam normal to the antennas, circularly polarized to the left. The patterns are chosen in each case to provide the closest possible average characteristics.

Figure C2: Improved anisotropy control through the morphogenetic approach enables more selective management of the radiated polarization state. The color bar maximum is adjusted according to the maximum gain obtained for the pair of metasurfaces, ensuring that the colors correspond to identical gain levels.

Figure C3: Under the simple conditions proposed, the morphogenetic approach enables an increase in gain and a widening of the bandwidth compared with the conventional approach.

It is important to point out once again that these comparisons have to be carried out under simplified conditions (in this case, uniform modulation of the anisotropy tensors), in neither case allowing optimized results to be obtained.

We are aware that a great deal of work remains to be done to fully position the advantages and disadvantages of this technique, which we did not set out to present as an ideal solution to all the limitations of conventional techniques, but rather as a promising avenue for breaking free from the uniform arrangements usually exploited.

We would like to thank you sincerely for the time and effort you have invested in studying our work, and we are convinced that your comments have helped to improve our work.

References

- [1] O. U. Uche, S. Torquato, and F. H. Stillinger, “Collective coordinate control of density distributions,” *Physical Review E*, vol. 74, no. 3, p. 031104, 2006.
- [2] S. Torquato and F. H. Stillinger, “Local density fluctuations, hyperuniformity, and order metrics,” *Physical Review E*, vol. 68, no. 4, p. 041113, 2003.
- [3] Y. Jiao, T. Lau, H. Hatzikirou, M. Meyer-Hermann, J. C. Corbo, and S. Torquato, “Avian photoreceptor patterns represent a disordered hyperuniform solution to a multiscale packing problem,” *Physical Review E*, vol. 89, no. 2, p. 022721, 2014.
- [4] A. Donev, I. Cisse, D. Sachs, E. A. Variano, F. H. Stillinger, R. Connelly, S. Torquato, and P. M. Chaikin, “Improving the density of jammed disordered packings using ellipsoids,” *Science*, vol. 303, no. 5660, pp. 990–993, 2004.

- [5] C. Gigli, Q. Li, P. Chavel, G. Leo, M. L. Brongersma, and P. Lalanne, “Fundamental limitations of huygens’ metasurfaces for optical beam shaping,” *Laser & Photonics Reviews*, vol. 15, no. 8, p. 2000448, 2021.
- [6] P. M. Piechulla, B. Fuhrmann, E. Slivina, C. Rockstuhl, R. B. Wehrspohn, and A. N. Sprafke, “Tailored light scattering through hyperuniform disorder in self-organized arrays of high-index nanodisks,” *Advanced Optical Materials*, vol. 9, no. 17, p. 2100186, 2021.
- [7] F. Chehami, C. Decroze, T. Pasquet, E. Perrin, and T. Fromenteze, “Morphogenetic design of self-organized correlated disordered electromagnetic media,” *ACS Photonics*, 2023.
- [8] N. A. Rubin, A. Zaidi, A. H. Dorrah, Z. Shi, and F. Capasso, “Jones matrix holography with metasurfaces,” *Science Advances*, vol. 7, no. 33, p. eabg7488, 2021.